# The Interplay between Bluetongue Virus Infections and Adaptive Immunity

**DOI:** 10.3390/v13081511

**Published:** 2021-07-31

**Authors:** Daniel Rodríguez-Martín, Andrés Louloudes-Lázaro, Miguel Avia, Verónica Martín, José M. Rojas, Noemí Sevilla

**Affiliations:** Centro de Investigación en Sanidad Animal, Centro Nacional Instituto de Investigación y Tecnología Agraria y Alimentaria, Consejo Superior de Investigaciones Científicas (CISA-INIA-CSIC), Valdeolmos, 28130 Madrid, Spain; rodriguez.daniel@inia.es (D.R.-M.); andres.louloudes@inia.es (A.L.-L.); miguelavia87@gmail.com (M.A.); mgarcia.veronica@inia.es (V.M.); rojas.jose@inia.es (J.M.R.)

**Keywords:** orbivirus, cytotoxic T-lymphocytes, T-helper cells, gamma-delta T-cells, ruminants, B-cells, T-cells

## Abstract

Viral infections have long provided a platform to understand the workings of immunity. For instance, great strides towards defining basic immunology concepts, such as MHC restriction of antigen presentation or T-cell memory development and maintenance, have been achieved thanks to the study of lymphocytic choriomeningitis virus (LCMV) infections. These studies have also shaped our understanding of antiviral immunity, and in particular T-cell responses. In the present review, we discuss how bluetongue virus (BTV), an economically important arbovirus from the *Reoviridae* family that affects ruminants, affects adaptive immunity in the natural hosts. During the initial stages of infection, BTV triggers leucopenia in the hosts. The host then mounts an adaptive immune response that controls the disease. In this work, we discuss how BTV triggers CD8^+^ T-cell expansion and neutralizing antibody responses, yet in some individuals viremia remains detectable after these adaptive immune mechanisms are active. We present some unpublished data showing that BTV infection also affects other T cell populations such as CD4^+^ T-cells or γδ T-cells, as well as B-cell numbers in the periphery. This review also discusses how BTV evades these adaptive immune mechanisms so that it can be transmitted back to the arthropod host. Understanding the interaction of BTV with immunity could ultimately define the correlates of protection with immune mechanisms that would improve our knowledge of ruminant immunology.

## 1. Introduction

Viral infections have long provided a platform to understand the workings of immunity. Early vaccination studies, dating back to the first immunizations in the 18th century by Edward Jenner that used cowpox lesion material to combat smallpox virus [1] and moving to the more systematic attenuation of the rabies virus performed by Louis Pasteur a century later [2], demonstrated that the immune system could be “trained” with less virulent pathogens to confer protection against these devastating diseases.

The study of pathogen infections has thus helped shape our understanding of multiple immunology notions. A prime example of this is the study of lymphocytic choriomeningitis virus (LCMV) infections that has revealed several immunology concepts (reviewed in [3,4,5,6,7]). For instance, studies of LCMV infections permitted the characterization of antigen recognition by T-cells [8,9,10] or understand memory development in T-cells [11,12]. Because the different strains of LCMV produce a wide spectrum of immune responses, ranging from acute viral clearance to chronic infections, studying this viral infection has also shed some light on viral induced immune dysfunctions. For example, LCMV “clone 13” persistent infection has been associated with the disruption of dendritic cell activity [13,14] and the induction of CD8^+^ T-cell exhaustion [15]. Thus, studying the immunosuppressive effects of viral infection on immunity allows us to understand regulatory immune mechanisms as well as routes of viral escape that can be targeted for therapy.

These concepts are highly relevant for economically important ruminant viral diseases as they can help develop better vaccination protocols. Ruminant viral pathogens can also provide us with the opportunity to better understand ruminant immunology. In the present review, we will discuss how the arbovirus bluetongue virus produces disease in domestic ruminants and how the host immune system responds to the infection. We review and present data on bluetongue virus effects on circulating lymphoid cells population and discuss how the virus evades adaptive immunity.

## 2. Bluetongue Virus: The Prototypical Orbivirus

Bluetongue virus (BTV) belongs to the *Reoviridae* family and is the prototypical orbivirus. BTV is an arbovirus, i.e., a virus transmitted to the mammalian host by arthropods. Infections by arboviruses represent an increasing challenge in global health as climate change has altered the distribution of competent vectors throughout the world. As a consequence, diseases that once were restricted to subtropical regions can now be transmitted endemically by these invading vectors. BTV is transmitted by the bite of *Culicoides* spp midges and affects wild and domestic ruminants [16]. The redistribution of competent vectors from Northern Africa to the whole Mediterranean basin, as well as the competence of some endemic *Culicoides* spp in transmitting the virus means that BTV can now be considered endemic in Europe Southern latitudes [17]. BTV produces a disease of mandatory notification to the World Organization for Animal Health (OIE) that mainly affects sheep. BTV infections lead to loss of productivity through increased mortality and abortion rates, reduced fertility, decreases in milk and wool yield as well as restriction to animal trade which account for an estimated 3 billion $ annual losses globally [18].

BTV is a dsRNA virus that possesses 10 segments encoding for 7 structural proteins (VP) and at least 4 non-structural proteins (NS) (Figure 1) [19]. The genetic material is encapsidated in a two-layered protein capsid. The outer capsid constituted by the VP2 and VP5 proteins contains the main antigenic determinants of neutralizing antibodies. As a result of VP2 high variability, multiple serotypes of BTV have been defined (at least 27 to date) [20,21]. Little cross-protection in terms of sterilizing immunity exists between serotypes, thus limiting the efficacy of current vaccination strategies based on serotype-specific inactivated vaccines [22]. Because of its segmented genetic material, BTV can form reassortants [23], which further complicates its control in regions where multiple serotypes circulate. The segmented genetic material, the RNA polymerase VP1, the RNA capping enzyme and methyl transferase VP4, and the RNA helicase VP6 are enclosed within the inner capsid constituted by VP7 and VP3 [19]. VP7 is one of the most conserved antigenic determinants between serotypes and as a result seroconversion assays are often based around the detection of antibodies against this protein [24]. BTV non-structural proteins are involved in virus replication and in impairing the host response to infection. NS1 promotes viral protein expression [25]; NS2 is an RNA binding protein which is the main component of viral inclusion bodies [26]; while NS3 is involved in virion egress [27,28,29]. VP3, NS3, NS4, and the putative NS5 are involved in impairing the host cellular response to BTV by acting as interferon antagonist and/or by promoting cellular shut-off [30,31,32,33,34,35].

## 3. The Control of BTV by Host Immunity

BTV infections in ruminants are often characterized by a biphasic viremia attributed to the IFN response [36,37]. The first viremia peak is likely controlled by the primary IFN response, but a second viremia peak is often detected once the IFN response subsides [36,37] and prior to the adaptive immune response taking place to clear the infection. Viremia in BTV infections can be long in some individuals and could be referred to as a state of semi-persistence [38,39]. This implies that the virus can effectively evade the effector arms of adaptive immunity, so that it can be transferred back to the arthropod host. Indeed, BTV has been shown to “hide” in erythrocytes, a mechanism that probably prolongs viremia [38].

Adoptive transfer experiments performed by Jeggo et al. helped establish that BTV protection is likely mediated by a combination of humoral and cellular immunity [40,41]. Transfer of neutralizing antibodies is effective at preventing re-infection from the same serotype, but confer limited protection towards different ones [42]. Jeggo et al. also showed that T cell adoptive transfer from BTV-recovered sheep protected from BTV challenge in monozygotic twins [41]. Cellular immunity in absence of neutralizing antibodies typically only provides partial protection [41,43], but it is cross-reactive between serotypes [44,45,46,47,48,49]. Indeed, vaccination based on BTV antigenic determinants that are conserved between serotypes (such as VP7 or NS1) have shown promising results in eliciting at least partial cross-serotype protection in animal models [44,46].

It is also important to mention that BTV infection severity varies greatly between susceptible species and even within the same host species [50]. Typically goat infection is asymptomatic, while cattle can develop moderate clinical signs in some cases. Sheep are more prone to develop clinically evident disease, but some animals can remain asymptomatic. This complicates BTV control as subclinical infected animals can act as reservoir for the disease. Control is even more troublesome because of the existence of wild ruminant hosts. Indeed, BTV is considered endemic in wild life in parts of North America and Africa [51,52], and studies in Europe have shown that wild ruminants populations are also infected during domestic outbreaks [53]. As such, deer and other wild ruminants such as mouflons or ibex could act as reservoir for the disease [54]. Differences in pathogenesis are also observed between BTV serotypes. For instance, the circulating BTV-8 strain responsible for the 2006 outbreak in Northern Europe produced frequent clinical infection in cattle [55].

## 4. Experimental BTV Infections of Natural Hosts and Prolonged Viremia

BTV infections are characterized by pyrexia, lack of appetite, and depression. In more severe cases, the disease manifests itself with conjunctivitis, congestion of the nasal and oral mucosa, and edema of the lips and face. These signs can develop to more prominent clinical features such as salivation mucopurulent nasal and oral discharge, ulcerations and hemorrhages in the nose lips and tongue, which sometimes progress to the cyanosis of the tongue that gave its name to the disease [24]. Scoring systems have been proposed to mark these signs [56,57,58,59], which can be useful to assess the degree of affectation in experimentally infected animals. Post-mortem histological evaluation can also be valuable to assess vascular lesions characteristic of the disease [60]. Most studies use temperature and clinical scoring of signs to macroscopically evaluate disease onset and progression [56,57,58,59]. In sheep experimentally infected with BTV-8, we were able to detect pyrexia and clinical signs such as nasal and oral congestion, depression, and loss of appetite which were scored based on the aforementioned systems [56,57,58,59] (Figure 2A,B).

In addition to the macroscopic evaluation of the signs, it is now accepted that quantitative PCR (qPCR) of reverse transcribed RNA samples obtained from blood is a good approximation of viral replication [62]. Typically, BTV infections are characterized by an early peak in replication which is controlled by the IFN system, followed by a secondary peak once the IFN response subside and before adaptive immunity clear the infection [36,37]. We detected viral RNA in blood samples as early as day 3 post-challenge, which then peaked at day 8 post-challenge (Figure 2C). A feature of BTV infections is that viremia is prolonged in some individuals, which is thought to favor the transmission back to the vector [38,39]. The mechanisms behind this prolonged viremia remains unclear.

Based on nucleotide and amino-acid sequence identities, BTV serotypes have recently been classified as “classical” (serotypes 1–24) and “atypical” for recent isolates which are difficult to serotype by traditional serological methods [63,64,65]. While “classical” BTV outbreaks are of compulsory notification to the OIE, infections with atypical BTV are not [66]. Most of these atypical BTV serotypes have been identified from asymptomatic goats and produce only mild clinical infections in sheep [63]. Infections with these atypical serotypes also appear restricted mostly to small ruminants. Recently, atypical BTV-25 appears to produce “persistent” infections, as a goat flock monitored over 4 years showed on and off PCR positivity for this serotype [67]. Interestingly, BTV-25 is only partially neutralized by antibodies in these animals [67]. It is unknown whether this state of BTV “persistence” is due to low levels of replication in the host or to cyclic reinfections by competent *Culicoides* spp. because of impaired virus clearance, or to a combination of both mechanisms. This also raises an important point for infections with “classical” BTV serotypes. “Classical” BTV serotypes are unlikely to produce persistence [68,69], but prolonged viremia is observable in some animals. Although viral RNA can be detected up to day 222 post-BTV inoculation in some cases, it is not thought that “classical” BTV can infect the feeding vector after day 21 post-inoculation both in cattle and sheep [70]. It thus appears that both classical and atypical BTV possesses, yet to be clarified, escape mechanisms that allow for prolonged viremia to occur. Understanding these escape routes will probably improve the control of the disease.

Seroconversion as assessed by the presence of antibodies reactive to the VP7 protein can occur early in infection [60], although antibody titers start to become evident from day 7 post-challenge and increase at later stage of the infection (Figure 2D). Some serotypes such as BTV-15 do have however different antibody induction kinetics whereby antibody titers to VP7 become apparent only at later stages of infection [71,72]. In BTV-8 reinfection experiments in sheep, VP7-expressing peripheral blood mononuclear cells are periodically detected by flow cytometry, although qPCR studies did not indicate that productive replication cycles were taking place as Ct values are stable once past the primary peak of replication [73]. BTV RNA levels were detectable in this study in spite of the presence of a concomitant cellular response to the BTV-VP7 protein [73]. Another study also showed in cattle that BTV-8 RNA could be detected in blood 60 days after reinfection in spite of the presence of neutralizing antibodies [72]. These reports illustrate that studies are still required to elucidate the mechanisms that allow BTV to prolong its circulation in spite of the presence of neutralizing antibodies and/or the existence of cellular immunity. BTV affinity for erythrocyte membrane could contribute to the prolonged detection of RNA in blood [38], it does however not explain the differences in viremia duration that are detected between animals. To the best of our knowledge, no studies have attempted to pinpoints which factors contribute to this prolonged high viremia that can be detected in some individuals. It is likely that, as for the IFN response that BTV overcomes in ruminants, the virus has developed mechanisms that have yet to be elucidated to prolong its replication and circulation in blood so that it can be transmitted back to susceptible *Culicoides* spp. First, it is nonetheless important to describe the interactions between BTV and immunity in order to later discuss BTV immune evasion mechanisms. The present discussion will be mostly centered on BTV effects on lymphoid cells with a particular focus on cellular immunity.

## 5. Effects of BTV Infection on Peripheral Lymphoid Cells Populations

### 5.1. BTV Infections Produce Leucopenia

BTV infection is known to induce leucopenia [73,74,75,76] typically within the first 2–7 days after experimental infections. This effect can precede the onset of fever in animals [77]. Alterations in leucocyte populations not only happen with blood circulating cells [74], but also in the lymph nodes after BTV infection [78]. BTV is known to be transported from the biting site by conventional dendritic cells (DC) to the draining lymph node where further replication likely occurs (Figure 3A) [79]. From there, the virus is systemically disseminated and infects further organs such as lungs or spleen. BTV can infect endothelial cells [80], cells of the monocytic phagocytic lineage, and lymphocytes [81,82]. Endothelial cell infections contribute to the characteristic hemorrhagic lesions produced by the infection [83], whereas infection of lymphocytes could account for some of the leucopenic events observed during the early stages of infection. Indeed, BTV infection induces apoptosis in peripheral blood mononuclear cells (PBMC) during the viremia peak which could contribute to leucocyte depletion [76]. This produces a transient immunosuppressive state that can render the animal susceptible to opportunistic infections [76].

Although BTV is considered to mostly produce acute infections, some animals can develop a chronic disease in which individuals suffer severe muscle loss and wool break. Some animals that appear to recover from the acute infection can also suddenly die, probably due to rapidly progressing pulmonary oedema [50]. Indeed, pneumonia and diarrhea signs in diseased animals are likely the result of opportunistic secondary infections [84] that take hold thanks to the transient immunosuppressive state produce by BTV infection. It is worth noting that the immunosuppressive period induced by BTV does not appear to be as prolonged or marked as in other ruminant viral diseases such as Peste des Petits Ruminants virus infections [85], in which the viral-induced acute immunosuppression leads to increased mortality in affected flocks due to secondary infections [86].

BTV has been described as pan-leucopenic, as primary replication has been reported in monocytes and neutrophils [77]. BTV has also been reported to affect lymphoid cell populations. Ellis et al. also showed decreases in CD4^+^ T cells, CD8^+^ T cells, and CD5^+^ CD4^−^ CD8^−^ T-cells (which would include γδ T-cells) following BTV-17 or BTV-10 infection in sheep [74]. They also showed that a similar reduction in these T-cell populations occurred in cattle during experimental BTV-17 infections [74]. Another report detected similar decreases in CD4^+^ T-cell and γδ T-cells population in BTV-1 and BTV-8 infections in sheep [87], which did not occur in vaccinated animals [88]. We studied by flow cytometry the variation in different peripheral blood mononuclear cell (PBMC) populations produced by BTV-8 infection (Figure 4A–F). We observed that by day 7 post-infection, the percentage of CD4^+^ T-cells (Figure 4A), CD8^+^ T-cells (Figure 4B), γδ T-cells (Figure 4E) and B-cells (Figure 4C) in PBMC decreased in BTV-8 infected sheep when compared to mock infected counterparts. This effect was concomitant with the peak of viremia and clinical signs in infected animals (Figure 2). Taken together, these reports are therefore indicative that the pan-leucopenic effect of BTV infection also involves alterations in lymphoid cell populations.

BTV infections appear to mostly affect CD4^+^ T cells and γδ T-cells populations in the periphery. It has been shown that γδ T-cells are recruited at the biting site of *Culicoides* spp midges [39]. Indeed, targeting of γδ T-cells by BTV has been suggested as a mechanism of overwintering in hosts [39]. In ruminants, γδ T-cell represent a significant proportion of T-cells that can reach 30% of PBMC in some adult animals [89]. This T-cell subset has been linked to responses to tuberculosis antigen in sheep and cattle [90,91], but may also play a role in antiviral responses in ruminants [92,93]. In spite of the relevance of this T-cell subset, little is known on the activity of γδ T-cells during BTV infections. Several reports [39,74,87] as well as the data herein presented concur to show that γδ T-cells may be targeted by BTV during the early stages of infection.

The effect of BTV infection on B-cells has been less studied. Sánchez-Cordón et al. reported no changes in B-cells population during the course of experimental infection with BTV-1 and BTV-8 [87]. In contrast, we are reporting here a prolonged decrease in the percentage of B-cells in infected sheep which lasted from day 5 to day 23 in a BTV-8 sheep infection (Figure 4C). BTV is known to disrupt B-cell responses by infecting follicular DC in the lymph node [94]. The decrease we observe in this experiment could indicate that BTV infects B-cells to limit their activity. Further work will be required to further elucidate this observation. NK cell percentage did not significantly change throughout the experiment (Figure 4F), indicating that lymphopenia was not sufficiently marked to trigger homeostatic NK cell expansion to compensate for cell loss [95]. This also indicates that BTV infection is not affecting this lymphoid cell population, although further analysis will be required to confirm this observation.

Overall, the pan-leukopenic effects of BTV infection appear to involve several lymphoid cell compartments. Indeed, the observation that apoptosis increased in PBMC as well as in spleen concurrent with the peak in viremia in BTV-23 infected sheep [76] suggests that BTV could be triggering lymphoid cell death and thus promoting an immunosuppressive state. Further work will be required to evaluate the exact contribution of this phenomenon to the pathogenesis of the disease.

### 5.2. BTV Infections Trigger Cellular and Humoral Immunity

Once the host overcomes the leucopenic phase of the infection, an adaptive immune response consisting in cellular and humoral components can be detected in infected animals (Figure 3B). As previously mentioned, Jeggo et al. showed in the 1980s using adoptive transfer experiments that both arms of adaptive immunity are required for BTV protection [40,41]. An increase in circulating CD8^+^ T-cells has been reported for experimental BTV infections in several studies using different BTV serotypes [47,73,74,76,87]. We also found a peak in circulating CD8^+^ T-cells at day 12 post-infection in the experiment described herein (Figure 4B). Barratt-Boyes et al. also showed in infected sheep that CD8^+^ T-cell numbers in efferent lymph from the lymph node (LN) proximal to the infection site increased from day 10 to 12 when compared to the efferent lymph of the contralateral LN [78]. This likely indicates that anti-BTV CD8^+^ T-cells are generated in the LN in response to the infection and that the expansion of these cells is also detectable in PBMC. Indeed, CD8^+^ T-cell responses in vaccinated animals have been associated with protection in the natural host [43,47]. Thus, it appears that CD8^+^ T-cell expansion forms an integral part of the adaptive immune response to BTV, which probably helps clear the infection.

Humoral immunity is also triggered by BTV infections (Figure 3B). Barratt-Boyes et al. showed that B-cells expand in the LN proximal to the BTV infection site in calves when compared to the contralateral LN of the same animal [78]. They also detected an increase in B-cells in efferent lymph from the proximal LN at day 10 to 12 [78]. We also detect an increase in CD21^+^ B-cells at day 15 to 23 in PBMC (Figure 4D). The complement receptor 2 (CD21) is a marker of mature B cells that promotes retention of immune complexes [96]. The increase in circulating CD21^+^ B-cells is concomitant to the increase in anti-VP7 IgG by day 22 (Figure 2D). It thus appears that in spite of the decreased percentage of circulating B-cells detected in the experiment herein described, the host can still mount an antibody response to BTV. Indeed, BTV is known to only produce a transient impairment of B cell responses, thus allowing the host to eventually mount a humoral response to the virus [94].

It should be noted that not all experimental infection studies give clear cut data when PBMC populations are analyzed [73,76,97] which is likely due to a combination of factors, such as isolate virulence, heterogeneity within breeds, susceptibility of breeds, and/or route and dose used for infection [98]. These alterations in PBMC populations after BTV infections are also likely to depend on the severity of the experimental infection. Overall, most studies concur to show that BTV affects multiple lymphoid populations in the early stages of infection which probably contribute to the pan-leucopenic effects of BTV infection. Ultimately, the ruminant host is capable of mounting a cellular and humoral immune response that helps clear the infection. Nonetheless, as previously discussed, BTV can prolong its circulation in the host so that it is transmitted back to the vector, which indicates that BTV possesses mechanisms that disrupt adaptive immunity. Some of the potential mechanisms are discussed in the final section of this review.

## 6. BTV Disruption of Adaptive Immunity

The mechanisms that allow BTV to prolong its viremia in spite of the presence of cellular and humoral immunity are poorly understood. Infective virus can circulate for at least 21 days in the host, while sensitive RT-qPCR techniques can pick up viral RNA presence for more than 200 days in some cases [70]. BTV is known to associate with erythrocytes [38] probably to facilitate its passage back to the *Culicoides* spp. Nonetheless, a mechanism of prolonged circulation based solely on the virus affinity for erythrocyte membrane is unlikely to fully explain the extended viremia particularly when cellular and humoral responses are elicited by the infection.

### 6.1. BTV Effects on Dendritic Cells

Broadly speaking, dendritic cells are professional antigen presenting cells that are essential for mounting adequate adaptive immune responses to pathogens (reviewed in [99]). Although these cells are part of the innate immune system, they represent a key link between innate and adaptive immunity. In their immature state, these sentinel cells can capture pathogen antigens in the periphery either through phagocytosis or even by becoming infected. They possess a myriad of pattern recognition receptors (PRR) that can recognize pathogen-associated molecular pattern (PAMP), such as viral genetic material. Once a PRR recognizes a PAMP, it triggers the activation of the DC which migrates to the proximal lymph node where it can present the pathogen antigens to naïve T-cells so that they can become activated. Similarly, specialized DC in secondary lymphoid organs (follicular DC) can capture and retain native antigen immune complexes so that antigen-specific B-cells can mature and differentiate to antibody-producing cells [100,101]. Unsurprisingly, viruses have evolved strategies that disrupt DC function and consequently the development of humoral and cellular immunity. For instance, disruption of T-cell antigen presentation is an integral part of LCMV immune evasion strategies [13,102], while measles virus targets multiple aspects of DC activation to promote T-cell hyporesponsiveness and immunosuppression [103].

Indeed, BTV is known to interact with DC during infection (Figure 3). It is now accepted that conventional DC (cDC) are responsible for transporting the virus from the biting site to the afferent LN where further viral replication can occur [79]. Once BTV reaches the LN, it can disrupt follicular DC activity [94]. BTV has the capacity to replicate in ruminants in presence of a concomitant antiviral IFN response (reviewed in [34]). We have shown that BTV presence in the LN triggered the expression of the IFN-stimulated gene Mx1 [94]. In spite of this, BTV could replicate and impair follicular DC activity resulting in delayed antibody response to a model antigen. Thus, BTV can target follicular DC to impair antibody responses.

The effect of BTV infection on cDC antigen presentation to T-cells is less well characterized. BTV is unable to productively infect sheep hematopoietic cells and thus could not impair in vitro differentiation of DC from hematopoietic precursors [104]. This resistance appears to be controlled by the host IFN response, since BTV can impair the differentiation of DC in type I IFN receptor-defective bone marrow precursors [104]. Upon infection, BTV increased the apoptosis and interfered with the upregulation of costimulatory molecules in these type I IFN receptor-defective DC [104], indicating that BTV-mediated impairment in IFN signaling could promote DC dysfunction. Hemati et al. showed that BTV is capable of productively infecting cDC in sheep [79]. Nonetheless, BTV-infected cDC are still capable of presenting BTV antigens to BTV-specific T cells obtained from the lymph of infected sheep [79]. Thus, BTV does not appear to affect cDC antigen presentation to antigen-experienced T-cells. Whether the virus affects cDC priming of T-cell responses remains to be elucidated.

### 6.2. BTV Effects on Humoral Immunity

There is some limited evidence that BTV can infect B-cells. Hemati et al. showed using NS2 immunofluorescence staining that some B-cells isolated from sheep lymph could be infected in vitro, although percentage of infected cells varied between BTV serotypes [79]. In contrast, another report using bovine PBMC indicates that B-cell infection by BTV is minimal [81]. The direct effect of BTV infection in B cells therefore remains an unanswered question.

As previously discussed, BTV can disrupt humoral responses by targeting follicular DC activity so that antibody responses are delayed (Figure 3C) [94]. This effect on the generation of antibody responses was still detected when the antigen was delivered 3 days post-infection showing that this mechanism probably provides BTV with a window of opportunity in which it can further its dissemination. In spite of the delayed antibody response produced by BTV infection, these antibody responses ultimately lead to the production of neutralizing antibodies [94]. Whether these response are long lived such as the ones elicited by sheep and cattle vaccination with inactivated virus that can be detected at least 7 years post-immunization remains to be determined [105,106]. It is also worth mentioning that IgG avidity to VP7 was impaired in sheep challenged with virulent BTV-8 as compared to animals that received an attenuated virus, indicating that BTV disruption of the antibody response could have long-term implication for BTV immunity [94].

### 6.3. BTV Effects on Cellular Immunity

The effects of BTV infection on cellular immunity are less well characterized. Infection of γδ T-cells at the biting site has been proposed as a mechanism for prolonged viremia. γδ T-cells can be persistently infected in vitro and are recruited at the *Culicoides* spp biting site in vivo [39]. Hemati et al. also detected infection in a small percentage of γδ T-cells isolated from sheep lymph [79]. Flow cytometry analysis of bovine PBMC also showed that activated γδ T-cells (IL-2 receptor^+^ cells) could become more readily infected in vitro than resting counterparts [81]. Overall, γδ T-cells could be a target for BTV during infection. The contribution of γδ T-cells to protective cellular immunity or pathogenesis is nonetheless unclear in BTV infections.

Two immunofluorescence analysis of CD4^+^ and CD8^+^ T-cell in vitro infection concur to show that CD4^+^ T-cells are more susceptible to infection than CD8^+^ T-cells in sheep and cattle [79,81]. In cattle, Barratt-Boyes et al. also showed that activated T-cells (IL-2 receptor^+^ cells) appear to be more susceptible to infection [81]. Further work will be required to better characterize the direct effects of BTV on T-cell activity, although infection of activated T-cells could represent an immunosuppressive mechanism of cellular immunity.

As previously discussed, CD8^+^ T-cell expansion is a common feature of BTV infections by days 10–15 (Figure 4B) [47,73,74,76,87] probably as a result of mounted anti-BTV T cell responses in the LN [78]. In spite of this expansion, primary anti-BTV responses, as assessed by IFN-γ production, are usually discreet at early timepoints. For instance, we only detected specific IFN-γ responses by ELISPOT in PBMC in 3 out of 8 infected sheep at day 9 post-challenge and in 2 out of 8 sheep at day 18 post-challenge (Figure 5). This is in line with previous work in which IFN-γ responses in PBMC became readily detectable upon reinfection [73]. It should be noted that these assays used inactivated virus as stimuli and thus response to non-structural proteins (such as NS1 which contain immunodominant epitopes [45,46]) are underestimated. Reinfection with BTV-8 produced a narrowing of the number of VP7 epitopes against which responses were detected [73]. This indicates that primary responses to BTV probably involve a broad repertoire of T cells, but that upon reinfection the cellular response tends to focus on a few dominant epitopes. Ellis et al. have also reported partial cellular response using proliferation assays with PBMC from BTV-17 infected sheep [74]. Two out of three sheep responded at day 7 and 14 post-infection by lymphoproliferation assays to purified BTV-17, although overall all three sheep responded at least at one time point. Calf PBMC responded similarly in these assays, indicating that ruminants can mount a cellular response to BTV. Interestingly, in all three sheep response to the T cell mitogen concanavalin-A was impaired by BTV infection when compared to day 0 samples [74]. This phenomenon was not observed with PBMC from infected calves, indicating that BTV may be more effective at impairing sheep cellular response.

It thus appears that BTV can induce a transient immunosuppressive state in T-cells (Figure 3C). The mechanism through which BTV impairs T-cell reactivity has yet to be established. Several putative mechanisms could be at play. Activated bovine T-cells are reported to be more susceptible to BTV infection, thus BTV could potentially eliminate these cells preferentially. Other viruses, such as LCMV or measles virus, manipulate DC activity to induce an immunosuppressive T-cell state [13,102,103]. Whether BTV also disrupts the priming of naïve T cell responses by DC remains an open question. It should also be noted that BTV-specific T-cells isolated from lymph produce IFN-γ as well as the immunomodulatory cytokine IL-10 upon antigen recognition [79]. IL-10 is known to be a key regulator of inflammation and cellular immunity that can be highjacked in the course of some viral infections (reviewed in [107]). Whether BTV manipulate this pathway remains to be determined.

Additionally, BTV is known to impair IFN-γ signaling [31]. IFN-γ is critical to the development of Th1 responses that target intracellular pathogens such as viruses [108]. The BTV-NS3 and -NS4 proteins have been shown to impair IFN-γ signaling in reporter assays [31]. Whether other BTV proteins that act as IFN antagonist, such as VP3 [35], are also involved in impairing IFN-γ signaling remains to be determined. It is therefore plausible that immune cells susceptible to BTV infection may therefore be unable to respond adequately to IFN-γ stimulation, which in turn could impair the antiviral Th1 response dependent on this cytokine [109]. Overall, although BTV effects on cellular immunity are not fully characterized, several studies hint at BTV capacity to also interfere with this arm of adaptive immunity. Much work still remains to be done to better understand T-cell responses to BTV and how they shape the course of infection.

## 7. Conclusions

Just as deciphering LCMV effects on immunity has helped elucidate essential immunology concepts, studying BTV infections will shed some light on the workings of the ruminant immune system. The study of BTV infections has already helped characterize new mechanisms of viral escape at the cellular and molecular level [31,94]. Many pieces of the puzzles are still nonetheless missing to fully comprehend BTV infections. For instance, the factors that govern the differences in susceptibility between hosts are still unknown. Thus, evaluating BTV effects on immunity could not only improve our understanding of ruminant immunology, but could also help untangle more general immunological mechanisms.

## Figures and Tables

**Figure 1 viruses-13-01511-f001:**
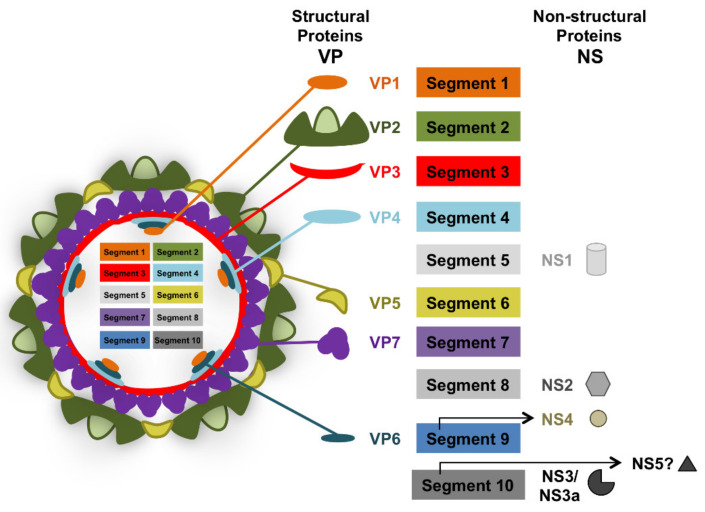
Schematic of bluetongue virus particle. BTV is a 10-segment dsRNA virus that belongs to the *Reoviridae* family. Its genetic material encodes for 7 structural proteins (VP) and 4–5 nonstructural proteins (NS). Segment 1 encodes for the RNA polymerase VP1. Segment 2 encodes for the highly variable outer capsid protein VP2. Segment 3 encodes for the inner core protein VP3. Segment 4 encodes for RNA capping enzyme and methyl transferase VP4. Segment 5 encodes for the nonstructural protein NS1 which forms cytoplasmic tubules during infection. Segment 6 encodes for the outer capsid protein VP5. Segment 7 encodes for the inner core protein VP7. Segment 8 encodes for the nonstructural protein NS2 which is part of the viral inclusion bodies. Segment 9 encodes for the RNA helicase VP6 and the non-structural protein NS4 which interferes with host immune response. Segment 10 encodes for the non-structural protein NS3 and its isoform NS3a which is involved in virion egress but also acts as an IFN antagonist. Segment 10 also putatively encodes for a fifth non-structural protein (NS5) that could be involved in host cellular shut-off.

**Figure 2 viruses-13-01511-f002:**
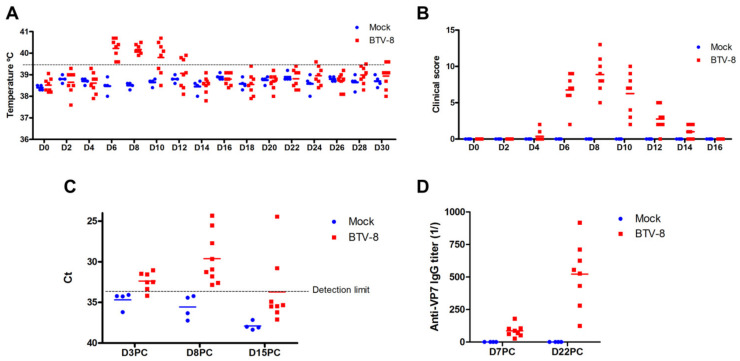
Assessment of BTV experimental infection in sheep: an example of BTV-8 infection in sheep. Sheep were experimentally infected intradermally with 2 × 10^6^ pfu BTV-8 (*n* = 8) or mock infected as control (*n* = 4). (**A**,**B**) BTV-8 infected sheep developed (**A**) fever and (**B**) characteristic clinical signs (pyrexia, apathy, loss of appetite, depression, oro-mucal lesions…) which were scored. (**C**) qPCR showed the presence of viral RNA in blood in 7 out of 8 infected sheep from day 3 post-challenge (D3PC). BTV RNA was detected in blood in all sheep at day 8 post-challenge (D8PC), while viremia was maintained up to day 15 post-challenge (D15PC) in 2 animals. (**D**) Seroconversion of BTV-8 infected sheep assessed by anti-VP7 ELISA. Anti-VP7 IgG are detectable at day 7 post-challenge (D7PC). Antibody titers increased by day 22 post-challenge (D22PC) in all infected sheep but titers are widely spread ranging from 1:125 up to 1:918. Detailed methodology can be found in Appendix A and in the following works [43,45,61].

**Figure 3 viruses-13-01511-f003:**
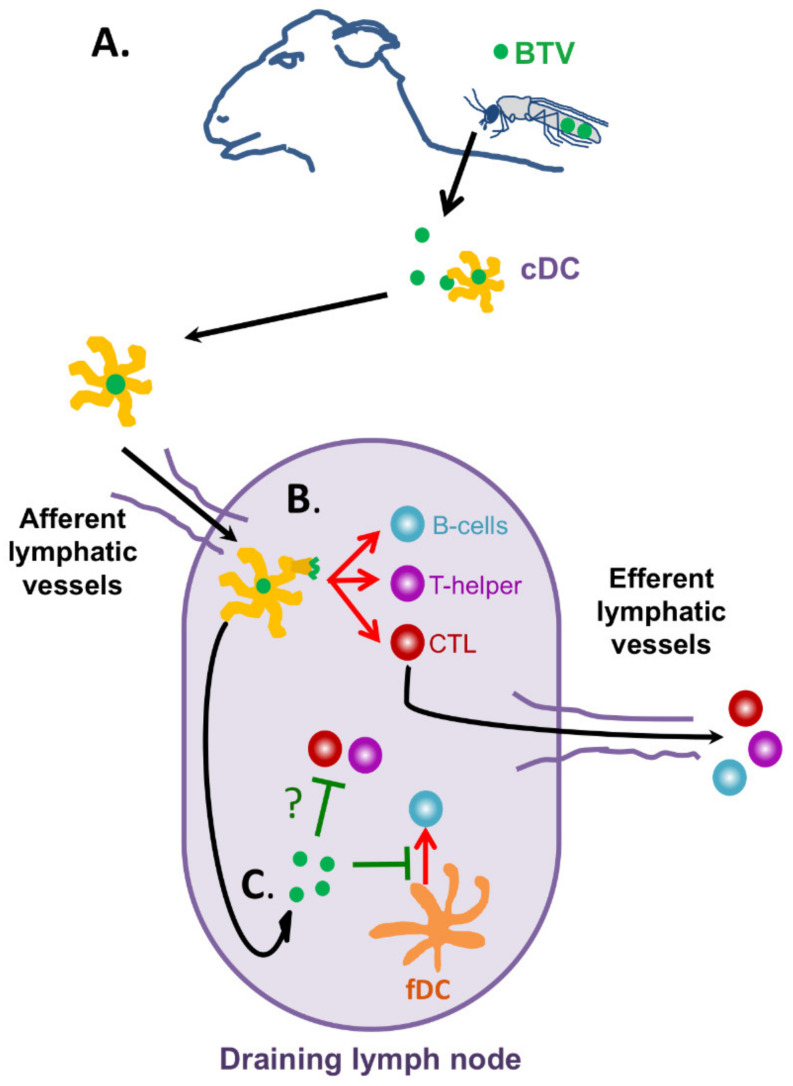
Overview of BTV effects on adaptive immunity. (**A**) BTV is transmitted by the bite of *Culicoides* spp midges to the ruminant host. The virus is captured by conventional dendritic cells (cDC), which transport it to the lymph node. (**B**) Once in the lymph node, cDC can deliver/present BTV antigens so that adaptive immunity is triggered. BTV is known to induce the activation of humoral immunity (B-cells) and cellular immunity (T-helper and cytotoxic T lymphocytes (CTL)). Both arms of adaptive immunity are required for protection from the virus. Once activated, B-cells, T-helper cells and CTL are released from the lymph node to help clear the viral infection. (**C**) BTV infection is known to impair adaptive immune responses. BTV can disrupt the function of follicular DC and consequently impairs antibody generation by B-cells. BTV also induces a transient hyporesponsiveness of T-cells to polyclonal stimuli. The mechanism of this effect is yet to be established (denoted as “**?**“ in the figure).

**Figure 4 viruses-13-01511-f004:**
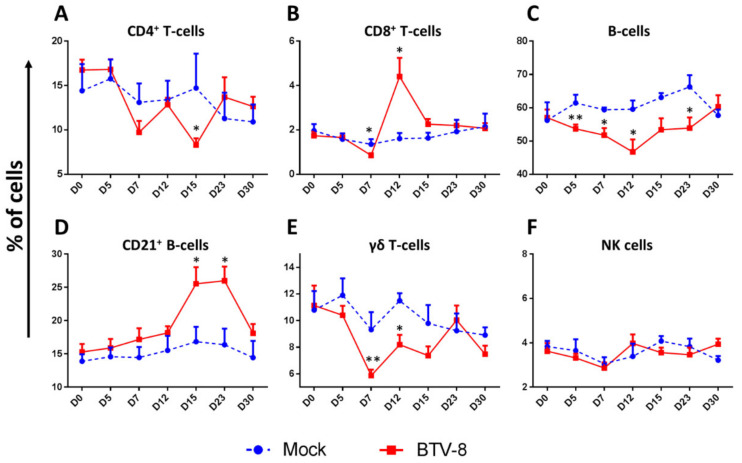
Flow cytometry analysis of lymphoid cell population in PBMC following BTV-8 infection. PBMC were obtained prior to infection (day 0), and at day 5, 7, 12, 15, 23, and 30 post-infection from BTV-8 or mock-infected sheep. (A–F) Flow cytometry analysis was performed to assess the percentage of CD4^+^ T-cells, CD8^+^ T-cells, B-cells, CD21^+^ B-cells, γδ T-cells, and NK cells in isolated PBMC. (**A**) CD4^+^ T-cell percentage decreased at day 7 post-BTV infection and was significantly lower at day 12 post-BTV infection when compared to mock-infected sheep. (**B**) CD8^+^ T-cell percentage decreased at day 7 post-infection and peaked at day 12 post-infection in BTV-infected sheep. (**C**) B-cell percentage decreased from day 5 up to day 23 post-infection in BTV infected sheep and only recovered to similar levels to mock-infected sheep by day 30. (**D**) Mature B-cells (CD21^+^ B-cells) percentage peaked in BTV-infected sheep at day 15 to day 23 post-infection when compared to mock-infected counterparts. (**E**) γδ T-cell percentage decreased by day 7 post-BTV infection and only recovered to levels similar to mock-infected animals by day 23 post-infection. (**F**) NK cell percentage did not appear to significantly change in BTV-infected sheep when compared to mock-infected animals. * *p* < 0.05; ** *p* < 0.01; Unpaired t-test (BTV-8 vs. Mock at indicated timepoint). Please refer to Appendix A for detailed methodology and Appendix A for gating strategy.

**Figure 5 viruses-13-01511-f005:**
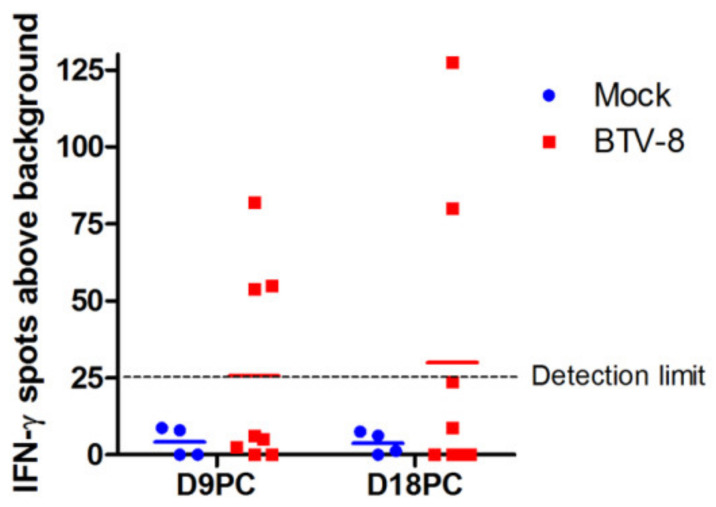
BTV-8 infection elicits anti-BTV IFN-γ producing cells. The presence of IFN-γ producing cells in sheep PBMC was assessed using ELISPOT assays. PBMC were stimulated with inactivated BTV-8 or mock cell lysate as negative control. All assays were performed at least in triplicates. Data are presented as average IFN-γ spots above background for 1 × 10^6^ PBMC at the indicated timepoint (D9PC: day 9 post-challenge; D18PC: day 18 post-challenge). The dotted line indicates the detection limit of the assay. IFN-γ producing cells to BTV-8 were detected in 3 out of 8 infected sheep at D9PC and in 2 sheep out of 8 infected sheep at D18PC. No specific IFN-γ responses were detected in PBMC from mock-infected sheep. Detailed ovine IFN-γ ELISPOT methods can be found in Appendix A and in [45].

## Data Availability

Data are available upon request.

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
