# Peer review of "The Interplay between Bluetongue Virus Infections and Adaptive Immunity"

_viruses, 2021, doi:10.3390/v13081511_

Round 1

Reviewer 1 Report

In their review article, the authors summarize the knowledge on the interaction between the adaptive immunity and BTV, an arthropod borne virus causing severe disease in infected sheep, while being less symptomatic in other ruminants. This is an important topic as BTV causes recurrent epidemics in different areas of the world, associated with important economic losses. To date, it is not clear why this virus can virtually infect any mammalian cell lines in vitro, but only ruminants in nature, with very different outcomes depending on the host. In this manuscript, the pathology of BTV in ruminants is well detailed, and it covers important aspects of the BTV adaptive immunity. My major criticisms are the long introduction on a virus (LCMV) different than BTV, the presence of several experimental data in a review article, and the plan of the manuscript that is sometimes confusing.

Major comments:

  1. The introduction and the first 27 references are almost only about the lymphocytic choriomeningitis virus. While it can be interesting, it seems off-topic, as this virus does not belong to the same family of BTV, and no parallel is provided with the literature on BTV. I suggest removing this part to increase the focus of your review.
  2. Regarding the presence of experimental data in a review article, I don’t think it is a problem as long as the data are mostly illustrating the results mentioned from the literature. In your manuscript, it is several experiments that would deserve to be in a research article. There as too much description of your results compare with the description of the literature, and this renders more as a discussion section of a research article.
  3. To me, it is essential to report the gating strategy of the flow cytometry data, at least as a supplementary figure. But again, I am not sure this is compatible with a review article.
  4. Your manus*cript plan is sometimes confusing to me. Section 6 is supposed to cover the adaptative immunity, but contains a paragraph on the cellular immunity. I would also add the paragraph 5.2 (“BTV trigger cellular and humoral immunity”) in section 6. Similarly, the paragraph 6.1 called “humoral immunity” contains information on IFN.
  5. Line 215, the authors mention the Peste des Petits Ruminants virus without much context. Could you detail how it is relevant to use it as a comparator with BTV? Similarly, BTV is briefly compared to LCMV line 696, without further context on the relevance of this comparison. If the authors want to provide context by comparing the BTV adaptive immunity to other viruses, which would be very interesting, then I suggest that they do it more systematically and with more detailed in the manuscript.

Minor comments:

  1. I would introduce a bit more information on the hosts of BTV. In particular, it would be interesting to mention line 139-145 other infected ruminants (wild animals such as deer).
  2. The paragraph on the cellular immunity response to BTV could be more detailed. In addition to NS3, there are other viral proteins that have been shown to act as immunomodulators, such as VP3 (interacting with the MAVS complex), or the NS4 protein.
  3. In figure 1, please add NS3a isoform
  4. Figure 2C, there is a discrepancy between the figure legend and the x-axis, is it 7 or 8 days post challenge?
  5. Line 246 need to be rewritten. “Similar to our data…”?

Reviewer 2 Report

The manuscript "The interplay between bluetongue virus infections and adaptive immunity" aims to "discuss how the arbovirus bluetongue virus produces disease in domestic ruminants and how the host immune system responds to the infection. We present data showing bluetongue effects on circulating lymphoid cells population and discuss how understanding the immune evasion mechanisms of this pathogen could improve our current knowledge of ruminant immunology".

Major comments:
The article seems a little disconnected, in part, due to the following reasons:
1- It is unclear why prolonging the comments regarding the LCMV virus in the introduction since it is a virus from a different viral family with different characteristics. The fact that viruses are useful to study the immune system is broadly known. 
2- The insertion of the new data provided by the authors is too extensive and moving the scope toward pathogenesis. Additionally, little detail is given about the methods used to generate the data, primarily given by citations in the supplementary file. Also, Fig. 2 seems unnecessary. 
3- Whereas the description of the lymphoid population during BTV infection was shown, I believe the review felt short in discussing the immune evasion mechanisms. Fragmented sentences with no references to substantiate the hypothesis are commonly found in the discussion.
4- P9 lines 293-301. Again, It is unclear why the immune responses to LCMV and BTV were compared. Different viruses, different families, different structures, and proteins. 
5- It would be interesting to describe further what is known about the interaction of individual BTV proteins with the immune system.

Minor comment:
1- What authors consider classical or atypical BTV strains. Is this based on genetic analyses? Phenotype? 

Round 2

Reviewer 1 Report

I thank the authors for their answers to my comments, and I am satisfied with the modified version of their manuscript.